**EMBO**
Molecular Medicine

# Inactivation of γ-secretases leads to accumulation of substrates and non-Alzheimer neurodegeneration

Hermien Acx[1,2,†], Lutgarde Serneels[1,2,†], Enrico Radaelli[1,2], Serge Muyldermans[3], Cécile Vincke[3], Elise Pepermans[1,2], Ulrike Müller[4], Lucía Chávez-Gutiérrez[1,2,*] & Bart De Strooper[1,2,5,**]

## Abstract

γ-Secretases are a family of intramembrane cleaving aspartyl proteases and important drug targets in Alzheimer's disease. Here, we generated mice deficient for all γ-secretases in the pyramidal neurons of the postnatal forebrain by deleting the three anterior pharynx defective 1 (Aph1) subunits (Aph1abc cKO Cre+). The mice show progressive cortical atrophy, neuronal loss, and gliosis. Interestingly, this is associated with more than 10-fold accumulation of membrane-bound fragments of App, Aplp1, Nrg1, and Dcc, while other known substrates of γ-secretase such as Aplp2, Lrp1, and Sdc3 accumulate to lesser extents. Despite numerous reports linking neurodegeneration to accumulation of membrane-bound App fragments, deletion of App expression in the combined Aph1 knockout does not rescue this phenotype. Importantly, knockout of only Aph1a- or Aph1bc-secretases causes limited and differential accumulation of substrates. This was not associated with neurodegeneration. Further development of selective Aph1-γ-secretase inhibitors should be considered for treatment of Alzheimer's disease.

**Keywords** Alzheimer's disease; Aph1 subunit; selective inhibition; side effects; γ-Secretase

**Subject Category** Neuroscience

## Introduction

γ-Secretases are a family of ubiquitously expressed membrane proteases consisting of four proteins: presenilin (PSEN), anterior pharynx defective-1 (APH1), nicastrin (NCT), and presenilin enhancer-2 (PEN-2) (De Strooper, 2003; Edbauer *et al*, 2003; Takasugi *et al*, 2003). Missense mutations in PSEN are the most

important cause of familial Alzheimer's disease (FAD) (Sherrington *et al*, 1995) (for an overview, see http://www.alzforum.org/mutations). These mutations affect the proteolytic processing of APP toward Aβ peptides. In contrast to general assumptions that their effects are quantitative, that is, increase Aβ$_{42}$ generation, the changes are mainly qualitative, shifting the spectrum of Aβ peptides toward longer versions that are apparently able to seed disease (Borchelt *et al*, 1996; Duff *et al*, 1996; Scheuner *et al*, 1996; Bentahir *et al*, 2006; Chávez-Gutiérrez *et al*, 2012; Szaruga *et al*, 2015; Veugelen *et al*, 2016).

In human, two different PSEN1 or PSEN2 and APH1A or APH1B variants can combine to generate four different enzyme complexes, whereas in rodents the *Aph1b* gene is duplicated, giving an additional *Aph1c* variant and resulting in 6 different rodent complexes (De Strooper, 2003; Hébert *et al*, 2004; Shirotani *et al*, 2004; Serneels *et al*, 2005). Alternative splicing of *PSEN* and *APH1* genes further adds to the complexity of the γ-secretase family (Gu *et al*, 2003; Hébert *et al*, 2004; Shirotani *et al*, 2004). The *Aph1b* and *Aph1c* genes in rodent are almost identical and located next to each other in the rodent genome; thus, we consider in our mouse experiments the *Aph1b* and *Aph1c* genes together as a homologue of the single human *APH1B* gene. The γ-secretase proteases cleave a broad spectrum of substrates, but it remains unclear whether specific substrates are cleaved by specific complexes (Kopan & Ilagan, 2004; Beel & Sanders, 2008; Jurisch-Yaksi *et al*, 2013; Sannerud *et al*, 2016). Cellular context is likely critical to understand the physiological functions of the different γ-secretases and their potential substrate selectivity (Sannerud *et al*, 2016). The recent failures of broad spectrum γ-secretase inhibitors in the clinic (Doody *et al*, 2013; Coric *et al*, 2015) dramatically illustrates our lack of basic knowledge of the γ-secretases (De Strooper, 2014).

Inactivation of either *Nct* or both the *Psen1&2* subunits in the excitatory neurons of the postnatal forebrain causes age-dependent neuronal loss, accompanied by astrocytosis and microgliosis without Aβ amyloidosis (Beglopoulos *et al*, 2004; Saura *et al*, 2004; Tabuchi *et al*, 2009; Wines-Samuelson *et al*, 2010). These

1 VIB Center for Brain and Disease Research, Leuven, Belgium
2 KU Leuven Department for Neurosciences, Leuven Institute for Neurodegenerative Disorders (LIND) and Universitaire Ziekenhuizen Leuven, University of Leuven, Leuven, Belgium
3 Cellular and Molecular Immunology, Vrije Universiteit Brussel, Brussels, Belgium
4 Institute for Pharmacy and Molecular Biotechnology (IPMB), University of Heidelberg, Heidelberg, Germany
5 UCL Dementia Research Institute (DRI-UK), London, UK
*Corresponding author. Tel: +32 16 37 69 35; E-mail: lucia.chavezGutierrez@cme.vib-kuleuven.be
**Corresponding author. Tel: +32 16 37 32 46; E-mail: bart.destrooper@cme.vib-kuleuven.be
†These authors contributed equally to this work

    

phenotypes are not observed when only *Psen1* is inactivated, but additional inactivation of one or two *Psen2* alleles causes neurodegeneration (Watanabe *et al*, 2014), indicating that a critical dose of γ-secretase activities is needed to maintain homeostasis of pyramidal neurons. The question has been raised whether the neurodegeneration in these knockout mice is relevant to processes occurring in Alzheimer's disease (Saura *et al*, 2004; Xia *et al*, 2016) or whether this reflects a non-typical neurodegeneration triggered by severe functional defects caused by the full knockout of all γ-secretases (Veugelen *et al*, 2016).

The loss of function of γ-secretases leads theoretically to the accumulation of their many substrates in the cell membrane. This has mostly been investigated in cell culture, and it is unclear to what extent this happens also *in vivo*, in the brain. It is also unclear whether the selective inactivation of one or more types of γ-secretases leads to differential effects on distinct substrates. If one hypothesizes that γ-secretase deficiency causes Alzheimer relevant neurodegeneration (Saura *et al*, 2004), this could theoretically be mediated by accumulation of unprocessed APP-carboxy-terminal fragments (CTFs) (Pera *et al*, 2013). Artificial overexpression of APP-CTFs is known to be neurotoxic *in vitro* (Yankner *et al*, 1989; Fukuchi *et al*, 1993), and *in vivo* (Neve *et al*, 1996; Oster-Granite *et al*, 1996). Inhibition of γ-secretase activity, but not β-secretase activity, causes synaptic and memory deficits in a mouse model of AD which is associated with accumulation of APP-CTF (Tamayev *et al*, 2012), suggesting that APP-CTFs, next to Aβ peptides, are also toxic species causing neurodegeneration. In the same line, chronic administration of a γ-secretase inhibitor, lowering Aβ but increasing APP-CTF, worsened cognitive function in mice overexpressing APP, while a "second generation" γ-secretase modulator that only affected Aβ production without changing APP-CTF levels, did not (Mitani *et al*, 2012). It should be noticed that the models used in these studies have strong overexpression of APP, and it remains to be seen whether such toxic APP-CTF level can be induced under physiological expression of App. It remains also unclear whether the accumulation of APP-CTFs is a general phenomenon in Alzheimer's disease (Pera *et al*, 2013) and how APP-CTFs would trigger or add to neurodegeneration, although various signaling pathways, some affecting neurite growth and dendritic arborization, have been proposed (Deyts *et al*, 2012).

Here we addressed some of these questions by genetically inactivating *Aph1a*- or *Aph1bc*-γ-secretases selectively in pyramidal neurons and by generating the triple *Aph1abc* knockout in these cells (*Aph1abc cKO Cre*⁺). We confirm that full knockout of all γ-secretases, as seen with the *Aph1abc cKO Cre*⁺, causes progressive neurodegeneration. Interestingly, this is not seen with the single *Aph1a* or *Aph1bc cKO Cre*⁺. The neurodegeneration in the *Aph1abc cKO Cre*⁺ mice was correlated with strong accumulation of several substrates, including a 20-fold increase in App-CTF levels, while the single cKOs showed only mild but differential accumulation of various substrates, demonstrating for the first time the *in vivo* selectivity of the two different γ-secretase subtypes. Finally, we find that deletion of the *App* gene does not modulate the neurodegenerative phenotype in the *Aph1abc cKO Cre*⁺ animals, indicating that App-CTFs are not necessarily involved in the mechanism(s) causing neurodegeneration in these animals.

# Results

## Loss of γ-secretase activity in CaMKIIa-positive neurons causes neurodegeneration

Here we fully inactivated γ-secretases by targeting the three *Aph1* genes in mouse pyramidal forebrain neurons. Mice homozygous for the floxed *Aph1* genes show already a depletion of the expression of the Aph1 subunits (Fig EV1) probably because of the insertion of the loxP sites. More importantly, the expression of other γ-secretase components is not affected (Fig EV1, quantified in Fig EV2A), and γ-secretase activity as evaluated by APP-CTF substrate accumulation or Aβ generation is not decreased in the *Aph1abc cKO Cre*⁻ condition (Fig EV2B and C). The mice do not show any phenotype when cortical thickness, neuronal cell counts per unit of sagittal length or inflammatory responses are evaluated at 9 months (Fig 1A and B). Furthermore, no major phenotypes have manifested over the 10 years that the colony has been maintained in our animal facility. For the next experiments, we consider therefore the mice with the floxed alleles as control, despite the lower expression of Aph1 subunits. We crossed them with mice that express Cre recombinase via the CaMKIIa promoter in order to delete the three *Aph1* genes in pyramidal neurons only (*Aph1abc cKO Cre*⁺). Western blots of cortical lysates show 30% less expression of Aph1a, while Aph1b expression was already undetectable in the Cre⁻ mice (Fig EV1). However, the effect of Cre-dependent deletion of the Aph1 components in the pyramidal neurons resulted in decreased Nct, Pen-2 and Psen1 expression levels by ± 50% (Fig EV2A). App-CTF levels were increased by 20-fold and Aβ$_{40}$ and Aβ$_{42}$ levels were both decreased by ± 35%, showing that γ-secretase activity in these neurons is strongly affected (Fig EV2B and C). The residual expression of γ-secretase components detected in the cortical lysates is explained by the neurons, glia and other cells that do not express Cre from the CaMKIIa promotor.

Immunohistochemical analyses of the brains of 3-month-old *Aph1abc cKO Cre*⁻ and *Aph1abc cKO Cre*⁺ mice (Fig EV4) show no difference in cortical thickness ($P > 0.99$), whereas at 6 months of age, serious cortical atrophy is seen (decrease of 10%; $P = 0.0009$) and the phenotype was further increased in 9-month-old *Aph1abc cKO Cre*⁺ mice (decrease of 22%; $P < 0.0001$) demonstrating that the genetic inactivation of all different γ-secretase complexes in pyramidal forebrain neurons leads to progressive cortical atrophy (Fig EV4). The loss of neurons in the *Aph1abc cKO Cre*⁺ mice was further confirmed by quantification of NeuN staining at 9 months ($799 \pm 41$ NeuN⁺ cells/mm cortical length versus $1{,}141 \pm 16$ NeuN⁺ cells/mm cortical length, *Aph1abc cKO Cre*⁺ versus *Cre*⁻) (Fig 1A and B). This was accompanied by increased Iba1 ($1.44 \pm 0.23\%$ versus $0.75 \pm 0.01\%$ Iba1-positive area) (Fig 1B) and Gfap ($13.56 \pm 0.75\%$ versus $2.04 \pm 0.90\%$ Gfap-positive area) (Fig 1B) immunoreactivity, diagnostic of microgliosis and astrogliosis, respectively.

## Single *Aph1* knockouts do not result in neurodegeneration

We have previously shown that depletion of Aph1b in mouse brain is sufficient to lower significantly Aβ generation in APP/PS1 mice (Serneels *et al*, 2009). Thus, we investigated the effects of the selective inactivation of the distinct Aph1-type enzyme complexes by

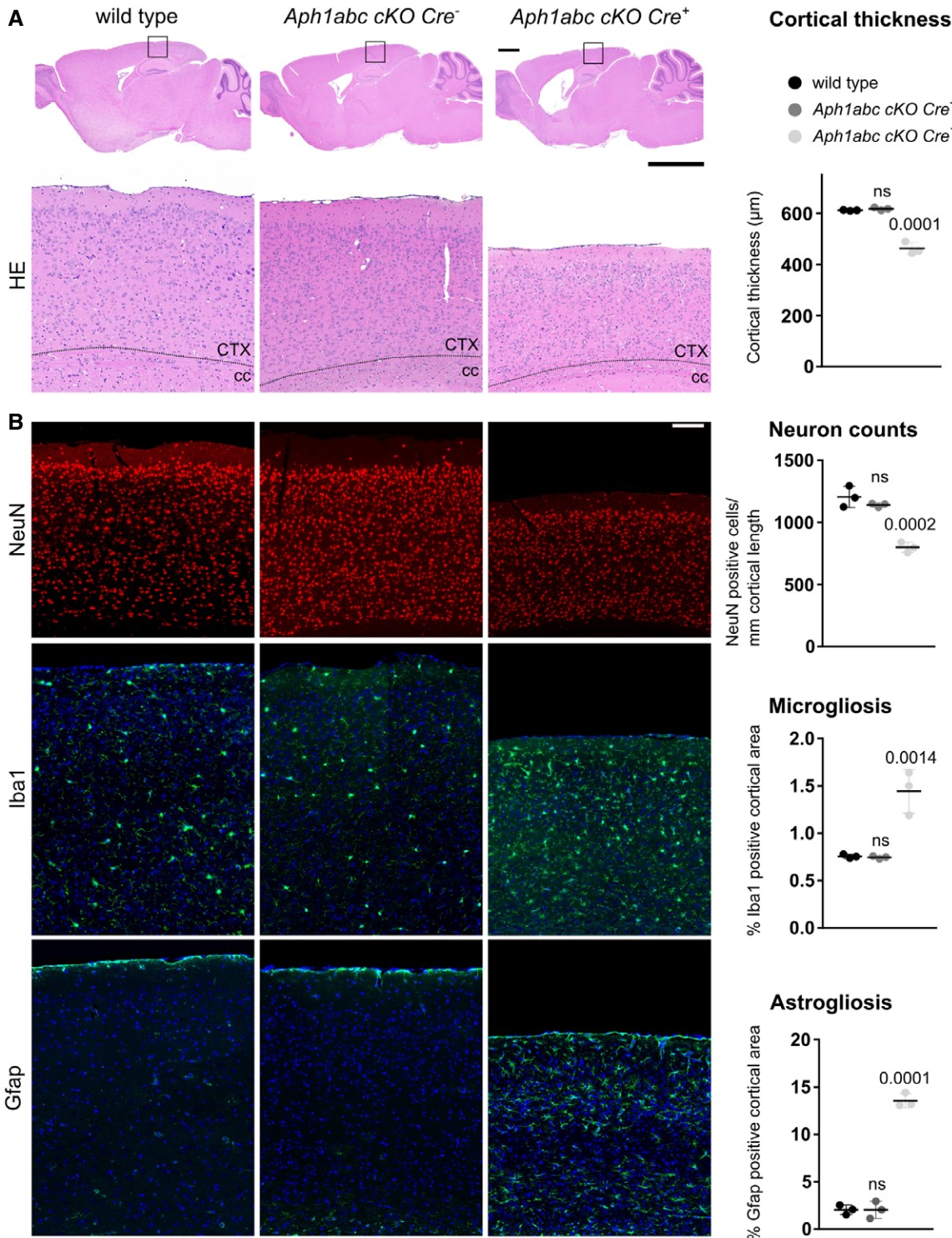

**Figure 1.  Loss of complete γ-secretase activity in CaMKIIa-expressing neurons leads to neurodegeneration.**

A   Hematoxylin/eosin staining of sagittal brain slices of 9-month-old mice. Upper panel scale bar = 1,100 μm; lower panel scale bar = 195 μm. Cortical atrophy is present in *Aph1abc cKO Cre+* mice, whereas absent in *Aph1abc cKO Cre−* mice.

B   NeuN, Iba1 and Gfap immunofluorescent labeling of parietal cortex of 9-month-old mice. Scale bar = 115 μm. Full cortical thickness counts of NeuN-positive cells, as determined via NeuN staining, was decreased in *Aph1abc cKO Cre+* mice. Reactive gliosis, as determined via Gfap and Iba1 staining, is present in *Aph1abc cKO Cre+* mice, whereas absent in *Aph1abc cKO Cre−* mice.

Data information: Mean, standard deviation, and *P*-values are shown. One-way ANOVA and Dunnett's *post hoc* test. CTX = cortex; cc = corpus callosum; ns = not statistically significant. *N* = 3.

generating mice with single *Aph1a* or single *Aph1bc* KO in the CaMKIIa-positive neurons. Remarkably, App-CTF is only accumulating in the brain of *Aph1bc cKO* Cre⁺ animals pointing toward an important role of Aph1bc-complexes in the processing of App-CTFs in these neurons (Fig EV3B and C). In line with this observation, Aβ levels are only decreased in the *Aph1bc cKO* Cre⁺ condition (Aβ$_{40}$ shows a trend for decrease, Aβ$_{42}$ is significantly decreased in the *Aph1bc cKO* Cre⁺ condition).

The current data suggest that Aph1bc is functionally more prominent than Aph1a at least in pyramidal neurons and with regard to the processing of App-CTFs. Interestingly and in contrast to the triple *Aph1abc cKO* Cre⁺ mice, the single *Aph1a cKO* Cre⁺ or *Aph1bc cKO* Cre⁺ mice do not show any signs of neurodegeneration, as demonstrated by the absence of atrophy of the cortex (Fig 2A), neuronal loss or inflammatory responses at 6 months of age (Fig 2B).

### Neurodegeneration in triple *Aph1* knockout mice is associated with massive accumulation of substrates

We analyzed to what extent C-terminal fragments of known γ-secretase substrates were accumulating in the different *Aph1 cKO* brains using Western blot. In the triple *Aph1abc cKO* Cre⁺ mice, we see strong accumulation of C-terminal fragments of App (18-fold), Aplp1 (16-fold), Nrg1 (13-fold), and Dcc (11-fold) and a milder accumulation of Aplp2 (1.5-fold), Lrp1 (2.4-fold), and Sdc3 (2.6-fold) (Fig 3A and C). Expression of the full-length proteins is not altered (Fig 3A). Remarkably, in the single *Aph1 cKO* Cre⁺ animals (Fig 3B and D) we see no or much more limited effects on, and differential alterations of the different substrates. For example, the C-terminal fragments of App, Aplp1, and Aplp2 are mildly accumulating in *Aph1bc cKO* Cre⁺ brains (1.7-, 2- and 1.7-fold, respectively), whereas they are unchanged in *Aph1a cKO* Cre⁺ brains. On the other hand, Sdc3 accumulates to a similar level in both *Aph1a* and *Aph1bc cKO* Cre⁺ brains, while Nrg1 and Lrp1 are accumulating in both, but slightly more in *Aph1a* than *Aph1bc cKO* Cre⁺ brains (3.5-fold versus 2.5-fold for Nrg1 and 1.6-fold versus 1.3-fold for Lrp1). These data demonstrate for the first time *in vivo* substrate selectivity of the two different γ-secretase subtypes. One should take into account that the changes only reflect what happens in pyramidal neurons.

### App is not involved in the neurodegeneration of the triple KO mice

We notice that one of the strongest accumulating substrates in the *Aph1abc cKO* Cre⁺ mice is App C-terminal fragment (18-fold) (Fig 3A and C). To examine the cellular pattern of App-CTF accumulation, we performed immune staining with an antibody that recognizes the C-terminus of App (Fig 4). Specificity of the staining is clear from the staining of *App KO* sections (Fig 4). The immunohistochemistry confirms entirely the strong signals we saw for App C-terminus in Western blot. Notably, while in the control brains the staining was mainly restricted to the cell bodies, in *Aph1abc cKO* Cre⁺ brains the staining for App increased in the neurites in CA3 and dentate gyrus regions of the hippocampus and in the parietal cortex overlying the hippocampus (Fig 4). As App-FL levels are unchanged in the *Aph1abc cKO* Cre⁺ lysates (Fig 3A), and App intracellular domain fragments (App-ICDs) cannot be generated

without active γ-secretase complexes, the increased staining must originate from the accumulation of App C-terminal fragments, as independently confirmed by Western blot analysis (Fig 3A).

We finally tested the hypothesis whether App-CTF accumulation could cause the neurodegeneration in the *Aph1abc cKO* Cre⁺ mice (Fig 5) by crossing them with the APP null mice. The loss of *App*, by itself, does not cause cortical neurodegeneration (Fig 5). Interestingly, loss of *App* did not modify the progressive cortical atrophy in the *Aph1abc cKO* Cre⁺, as evaluated by measuring the thickness of the neocortex (Fig 5A), neuronal count and reactive micro- and astrogliosis in 9-month-old mice (Fig 5B). Of notice, *App KO* mice displayed equal levels of NeuN, Iba1 and Gfap immunoreactivity compared to wild-type control mice, showing that depletion of *App*, on its own, has also no effect on neuronal survival and gliosis (Fig 5B). Altogether, these results demonstrate that the accumulation of App-CTFs is not the cause of the neurodegeneration observed in *Aph1abc cKO* Cre⁺ mice.

## Discussion

Here we demonstrate that the combined inactivation of the three *Aph1* subunits in postnatal pyramidal neurons in the mouse brain leads to a similar neurodegenerative phenotype as previously seen with *Nct* and *Psen1&2* knockouts in these cells (Saura *et al*, 2004; Tabuchi *et al*, 2009; Wines-Samuelson *et al*, 2010). Importantly, we do not observe this progressive neurodegeneration in single *Aph1a* or *Aph1bc cKO* Cre⁺ mice. The data thus support the possibility that different γ-secretases can at least partially compensate for each other's loss.

Introducing LoxP sites into the genes encoding the different Aph1 proteins caused on its own a strong decrease in their protein expression (Fig EV1). This partial loss of function mutation (hypomorphic mutation) had little effect on the stability of the other γ-secretase subunits (Figs EV1 and EV2A), and on γ-secretase enzymatic activities as deduced from App-CTF and Aβ levels (Fig EV2B and C). This claim is further corroborated by the lack of neurodegeneration, neuronal loss, astrogliosis or microgliosis in these mice (Fig 1). With the expression of Cre driven by the CaMKIIa promotor we expect that the loxP sites will recombine, resulting in complete abrogation of Aph1 expression in pyramidal neurons of the forebrain. Other neurons and glia cells continue expressing γ-secretases normally, and therefore, Western blots of cortical extracts are providing only a crude estimate of the effects (Figs EV1 and EV2A). However, App-CTF and other substrates accumulate and decreased Aβ generation is readily observed (Fig EV2B and C). The progressive neuronal loss and progressive astro- and microgliosis suggest that γ-secretase activity is essential in these neurons for their survival (Fig 1). The mechanism(s) underlying the neurodegenerative process remain unclear, but the age dependency and the progressive nature of the disorder in the mice (Fig EV4) have led to speculations that full γ-secretase knockout reflects what occurs in FAD patients (Saura *et al*, 2004; Xia *et al*, 2016). Our data suggest however underlying mechanisms that are quite different from the processes causing AD. First, the full and indiscriminate inactivation of γ-secretases in pyramidal neurons leads to huge accumulations of various substrates like App, Aplp1, Nrg1 and Dcc, while Aplp2, Lrp1 and Sdc3 accumulate to lesser extents (Fig 3). To our

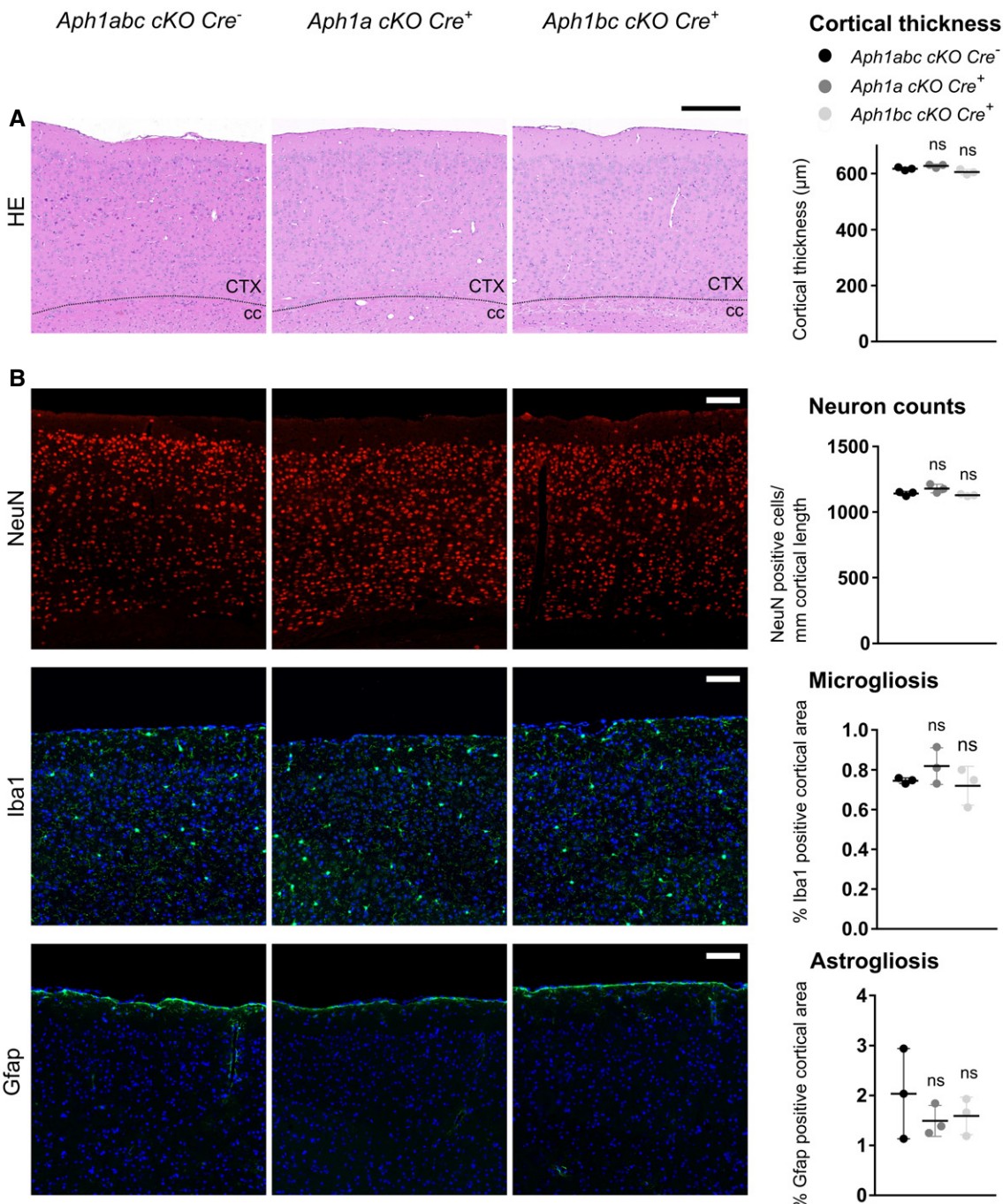

**Figure 2.  No neurodegenerative phenotype observed in single *Aph1a* and *Aph1bc cKO Cre⁺* mice.**

A   Hematoxylin/eosin staining of sagittal brain slices of 6-month-old mice. Scale bar = 195 μm. Cortical atrophy is absent in *Aph1a cKO Cre⁺* and *Aph1bc cKO Cre⁺* brains.

B   NeuN, Iba1, and Gfap immunofluorescent labeling of parietal cortex. Scale bar = 115 μm. Full cortical thickness counts of NeuN-positive cells, as determined via NeuN staining, was comparable in control, *Aph1a cKO Cre⁺*, and *Aph1bc cKO Cre⁺* mice. Reactive gliosis, as determined via Gfap and Iba1 staining, is absent in *Aph1a cKO Cre⁺* and *Aph1bc cKO Cre⁺* mice.

Data information: Mean and standard deviation are shown. One-way ANOVA and Dunnett's *post hoc* test. ns = not statistically significant. *N* = 3.

knowledge, such strong accumulations of membrane-bound protein fragments have never been observed in AD patients. Second, the phenotype in the triple KO mouse was not affected by the cross with *App KO* mouse, which deletes the strongly accumulating App-CTF

(Fig 5). The hypothesis that the neurodegeneration in the full γ-secretase knockout animals is AD related (Saura *et al*, 2004; Tabuchi *et al*, 2009; Xia *et al*, 2016) is difficult to maintain if APP does not modify this phenotype. Finally, as we show here,

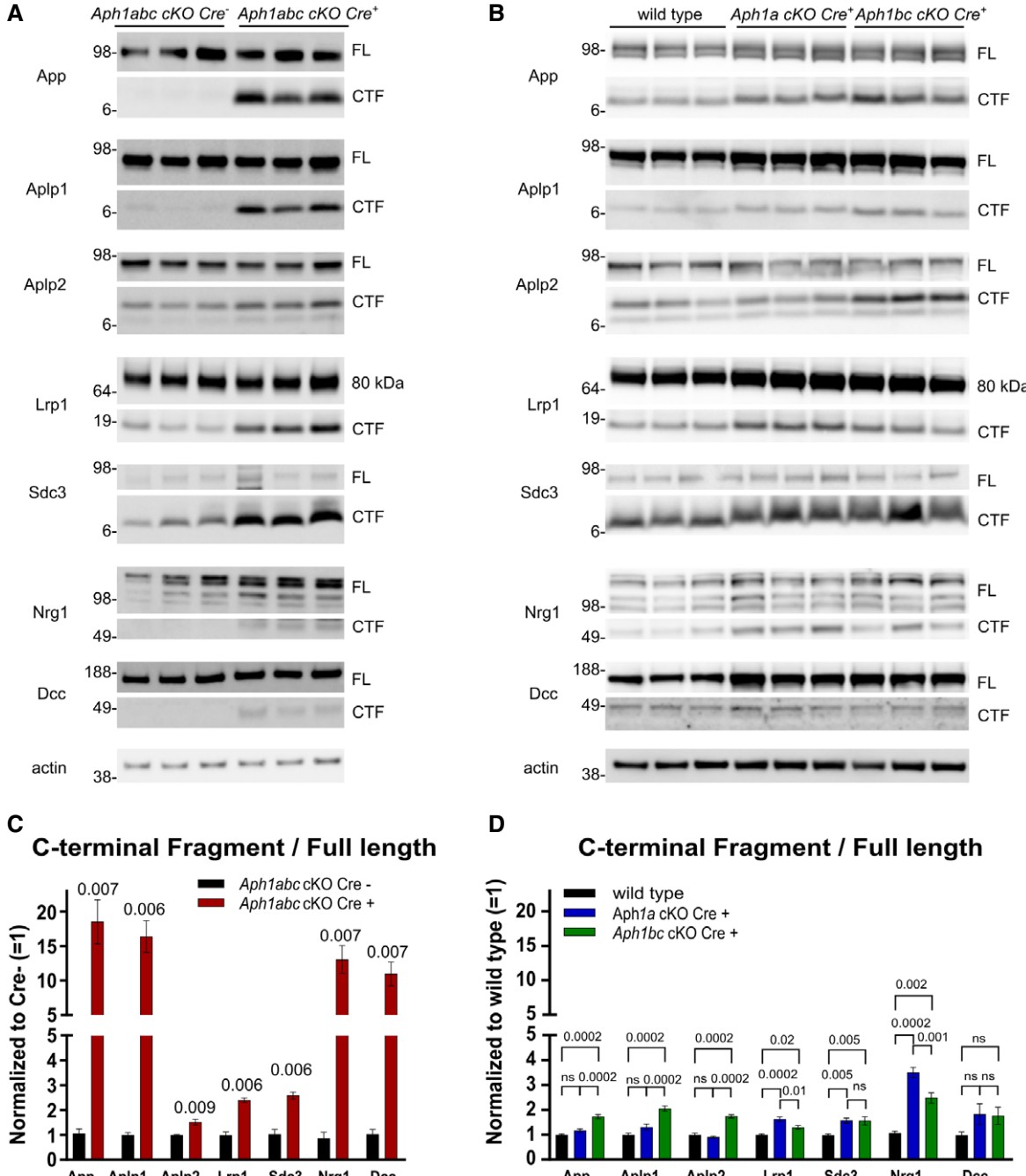

**Figure 3. Differential accumulation of substrates *in vivo*.**

Hippocampal lysates were analyzed by immunoblotting using antibodies against the C-terminus of the different γ-secretase substrates. For every substrate, full-length (FL) or the 80-kDa furin-cleaved LRP fragment and C-terminal fragments (CTF) are shown. β-Actin was used as loading control to normalize the data. Molecular weight markers are depicted in kDa.

A Western blot analysis of hippocampal lysates from *Aph1abc cKO Cre⁻* and *Aph1abc cKO Cre⁺* mice, three representative samples are shown. Five mice per genotype were analyzed.

B Western blot analysis of hippocampal lysates from wild-type, *Aph1a cKO Cre⁺*, and *Aph1bc cKO Cre⁺*, three representative samples are shown. Ten mice per genotype were analyzed.

C FL and CTF protein levels in panel (A) were quantified and FL/CTF ratios are plotted, normalized to *Aph1abc cKO Cre⁻* controls. Significant differences in protein expression between controls and *Aph1abc cKO Cre⁺* mice were assessed using Student's *t*-test per protein (GraphPad), followed by false discovery rate (FDR) *P*-value adjustments (R, version 3.3.1) to correct for multiple protein testing. Mean, SEM, and *P*-values are shown, ns = not statistically significant, *N* = 5.

D FL and CTF protein levels in panel (B) were quantified and FL/CTF ratios are plotted, normalized to wild-type controls. Differences in protein expression between wild-type, *Aph1a cKO Cre⁺*, and *Aph1bc cKO Cre⁺* were computed using one-way ANOVA per protein, followed by FDR *P*-value adjustment of the Tukey's *post hoc* test. Mean, SEM, and *P*-values are shown, ns = not statistically significant, *N* = 10.

Source data are available online for this figure.

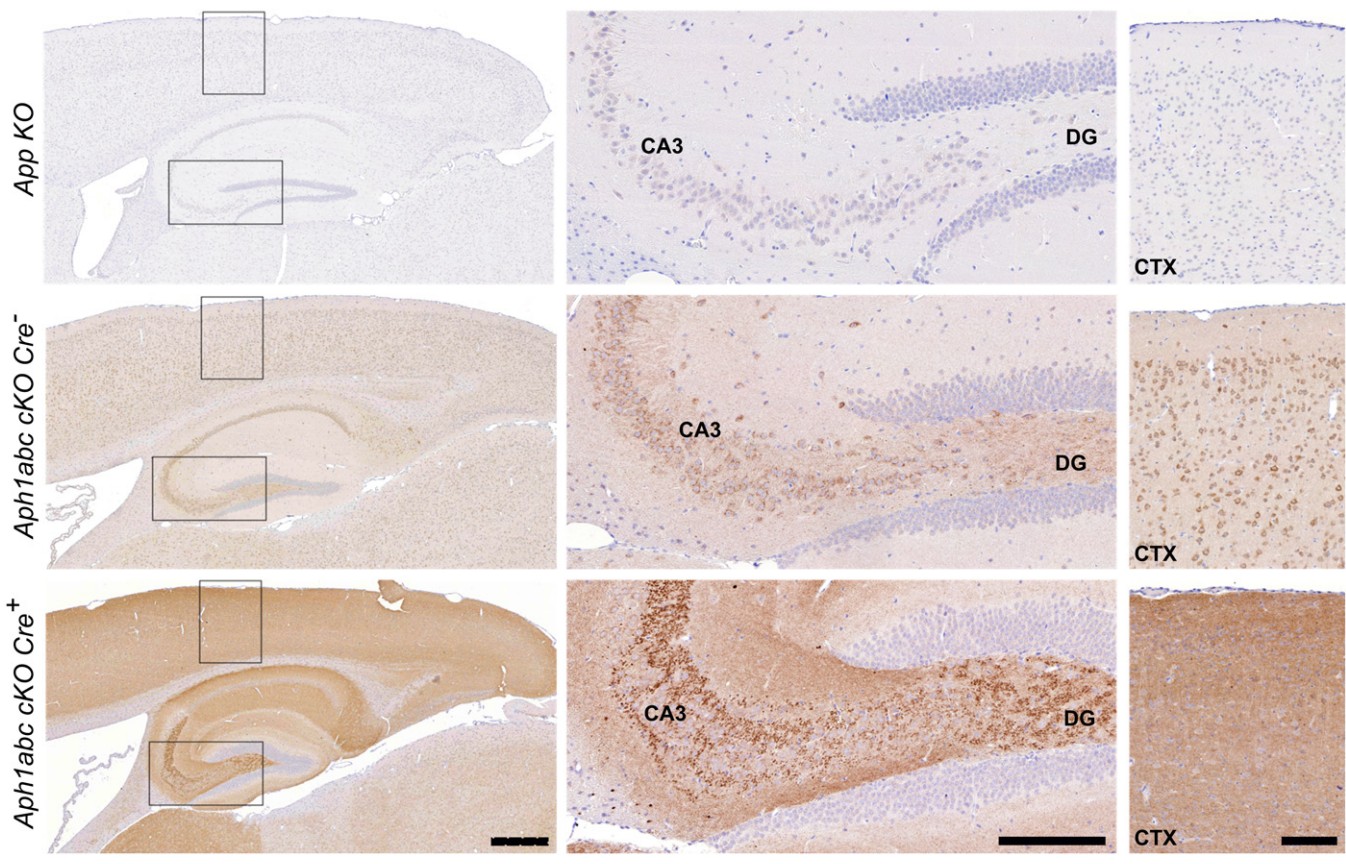

**Figure 4.  App-CTFs accumulate in the neurites and synaptic compartments in *Aph1abc cKO Cre⁺* brains.**
Immunostaining with an antibody recognizing the C-terminal part of App. Sagittal brain slices from *App KO* mice were used to show the specificity of the antibody. Conditional deletion of the *Aph1* genes in the pyramidal neurons causes accumulation of App-CTFs in the neurites in the hippocampal and cortical areas. Zoom-ins on the dentate gyrus (DG) and CA3 region of the hippocampus and the parietal cortex overlying the hippocampus show that expression of App and App-CTFs is mainly confined to the neuronal somata in control brains, whereas App-CTFs accumulate in the neurites and synaptic compartments in *Aph1abc cKO Cre⁺* brains. Scale bar for left panels = 390 μm; scale bar for central panels = 150 μm; scale bar for right panels = 110 μm.

neurodegeneration is only observed when all 6 γ-secretases are inactivated together in the mouse neurons (Figs 1 and 2). Such situation (with four human γ-secretases all inactivated) is highly unlikely to occur in heterozygous FAD patients carrying partial loss of function mutations (Szaruga *et al*, 2015; Veugelen *et al*, 2016).

As the accumulation of App-CTFs by itself is not causal to the neurodegeneration in the triple KO mice, two other hypotheses remain to be further investigated. First, it could be that the neurodegeneration is a consequence of the huge accumulation of various γ-secretase substrates in the cell membrane of the neurons. Overcrowding of the membrane with transmembrane domains of various proteins is not unlikely to cause havoc in neurons in a non-specific way. The alternative possibility remains that the neurodegeneration is caused by deficient processing of a specific substrate crucially important for intracellular signaling and maintenance of neuronal homeostasis. App (as shown here) and Notch1&2 (Zheng *et al*, 2012) are already excluded. It is clear that further identification and validation of endogenous substrates of the different γ-secretases in the adult brain is of great importance to understand better the biological role of these intriguing proteases (De Strooper, 2014). For instance, and highly relevant to AD therapy, the way the non-selective γ-secretase inhibitors semagacestat (Doody *et al*,

2013) and avagacestat (Coric *et al*, 2015) were dosed in the failed clinical trials has likely caused short but complete blocks of all enzymes at the same time (discussed in De Strooper, 2014). This repetitive complete blockage of all γ-secretases mimics in a pharmacological way the situation in the *Aph1abc cKO Cre⁺* mice. Although we did not perform pharmacological experiments to investigate this further, we speculate that intermittent accumulations of transmembrane domain fragments in neurons might have contributed to the aggravated cognitive performance of the treated patients.

The most important and hope giving result of the current work is the very mild effects of single *Aph1a* or *Aph1bc* knockouts in the brain of adult mice (Figs 2 and 3). Analysis of the accumulation of different substrates shows that quantitative effects on substrate levels remain relatively small with maximum a doubling of the normal levels of specific substrates (to be compared with the 10- to 18-fold changes seen with some substrates in the full *Aph1abc cKO Cre⁺* animals). Remarkably, the *Aph1bc cKO Cre⁺*, even when limited to the forebrain pyramidal neurons, has a significant effect on Aβ₄₂ generation (Fig EV3C) confirming previous findings with the *Aph1bc* full knockout which could prevent amyloidosis and cognitive decline in an APP overexpressing AD model (Serneels *et al*, 2009). Of note, a certain substrate specificity of the

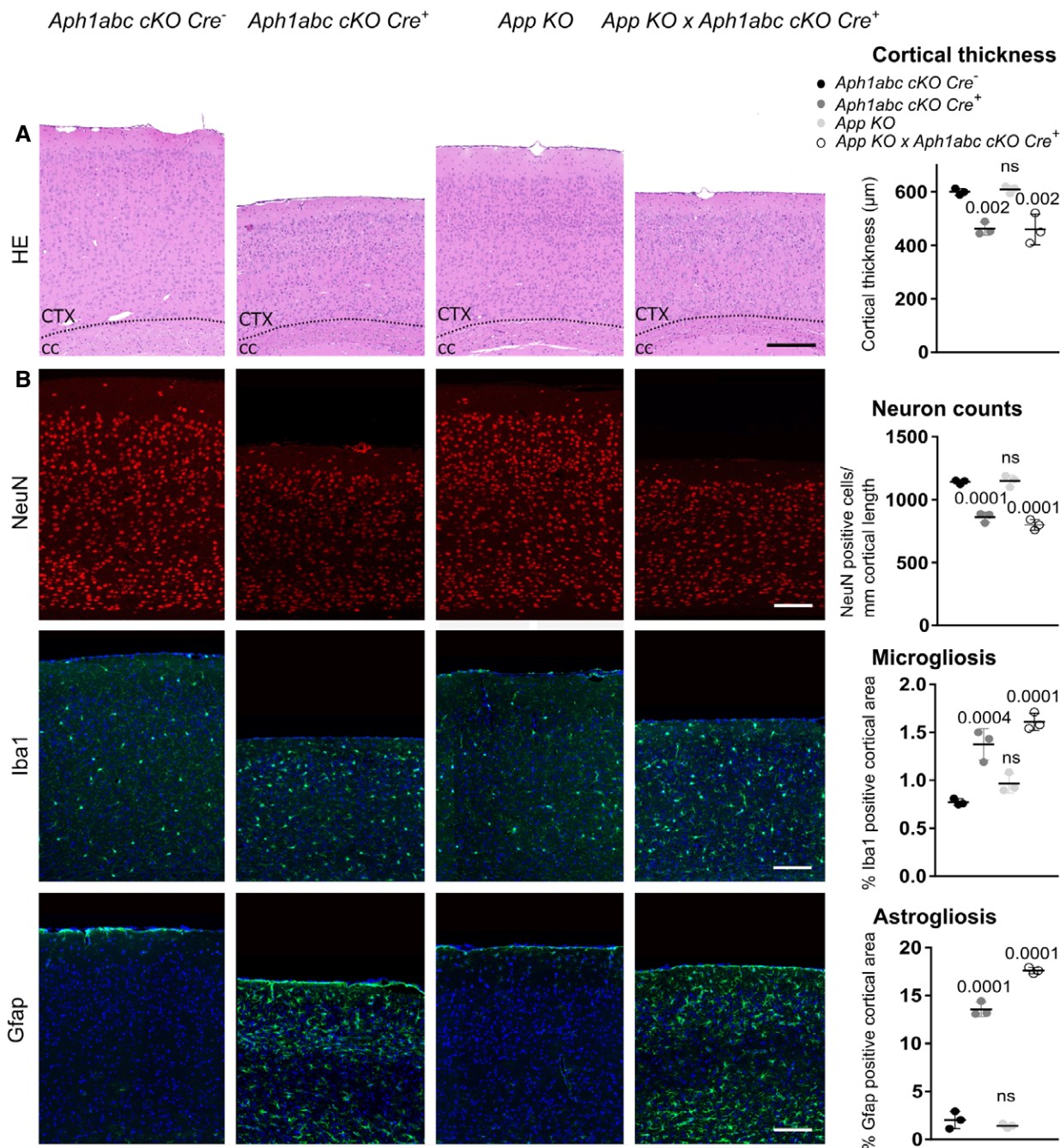

**Figure 5.  *App* depletion does not rescue cortical atrophy and gliosis in *Aph1abc cKO Cre⁺* mice.**

A   Hematoxylin/eosin staining of sagittal brain slices of 9-month-old mice. Scale bar = 195 μm. Progressive cortical atrophy is present to the same extent in *Aph1abc cKO Cre⁺* and *Aph1abc cKO Cre⁺* × *App KO* mice, whereas absent in control and *App KO* mice.

B   NeuN, Iba1, and Gfap immunofluorescent labelings of parietal cortex. Scale bar = 115 μm. Full cortical thickness counts of NeuN-positive cells, as determined via NeuN staining, was comparable in control and *App KO* mice, whereas decreased to the same extent in *Aph1abc cKO Cre⁺* and *Aph1abc cKO Cre⁺* × *App KO* mice. Reactive gliosis, as determined via Gfap and Iba1 staining, is present to the same extent in *Aph1abc cKO Cre⁺* and *Aph1abc cKO Cre⁺* × *App KO* mice, whereas absent in control and *App KO* mice.

Data information: Mean, standard deviation, and *P*-values are shown, ns = not statistically significant. One-way ANOVA and Dunnett's *post hoc* test. *N* = 3.

γ-secretases in function of their Aph1 subunits is observed: App, Aplp1 and Aplp2 are mildly accumulating in *Aph1bc cKO Cre⁺* brains, but not in *Aph1a cKO Cre⁺* brains, while Nrg1 and Lrp1 are accumulating slightly more in *Aph1a* than *Aph1bc* cKO *Cre⁺* brains (Fig 3). Differential expression and co-localization of substrates and enzyme complexes at the cellular or even subcellular level

(Sannerud *et al*, 2016) are contributing to this, but also distinctive intrinsic enzymatic properties of the different complexes (Mastrangelo *et al*, 2005), in which the Aph1 proteins act as allosteric subunits, may account for differential processing of substrates (Acx *et al*, 2014).

Altogether, we show that the loss of the three *Aph1* subunits in pyramidal neurons of the postnatal forebrain leads to a progressive neurodegenerative phenotype accompanied by the severe accumulation of γ-secretase substrates. Depletion of App did not modulate cortical atrophy, neuronal loss or inflammation, which suggests that the resulting phenotype is not typical for AD. Since γ-secretase substrate accumulation is limited in single *Aph1a* or *Aph1bc* cKO $Cre^+$ mice, and not associated with a neurodegenerative phenotype, further development of selective γ-secretase inhibitors, and in particular drugs targeting the combination PSEN1/APH1B, might be considered for treatment of Alzheimer's disease. A word of caution is indicated, however. First, we analyzed mice only at biochemical and microscopic-morphological levels, and we can thus not make any statements with regard to potential behavioral or electrophysiological abnormalities in the different mice investigated here. Second, it should be stressed that we here only inactivate γ-secretases in neurons that express Cre driven by the CaMKIIa promotor. Pharmacological inhibitors would block also γ-secretases in other cells and peripheral tissues, and therefore, further scrutiny is needed. We previously generated full Aph1bc knockouts and have shown that these mice are healthy and show very little problems apart from a deficit in operational memory (Dejaegere *et al*, 2008; Serneels *et al*, 2009).

# Materials and Methods

## Generation of mice

Mice conditionally targeted for *Aph1a*, *Aph1bc* or *Aph1abc* were previously generated (Serneels *et al*, 2005). cKO mice were generated by crossing with Tg (CaMKIIa-*Cre*)$^{T291STl}$ mice (Jackson laboratory) resulting in knockout of the targeted genes in the postnatal pyramidal neurons of the forebrain (Mayford *et al*, 1996; Yu *et al*, 2001). The *Aph1a*, *Aph1bc* and *Aph1abc* cKO animals were heterozygous for the CaMKIIa-*Cre* transgene while Cre-negative littermates were used as controls. App$^{tm2Cwe}$ (in this study called *App KO*) animals were previously generated (Li *et al*, 1996). Simple crossings led to the *Aph1abc cKO × App KO* animals. All colonies were kept on an inbred C57Bl/6J background. Both females and males were included in the study. The morphological and biochemical analyses were done on different time points (3, 6, or 9 months of age); this is mentioned in the respective figure legends. Mice are housed in cages enriched with wood-wool and shavings as bedding, given access to water and food *ad libitum*. All experiments were approved by the Ethical Committee on Animal Experimenting of the University of Leuven (KU Leuven).

## Antibodies

Rabbit polyclonal antibodies against Psen1-NTF (B19), Aph1a (B80.3), Nct (9C3), Pen-2 (B126), and App C-terminus (B63.3) have been described (Annaert *et al*, 2001; Esselens *et al*, 2004). Aph1bc (L82) was generated in the laboratory by immunizing rabbits with a QDKNFLLYNQRSR peptide. Aplp1 (W1CT) and Aplp2 (W2CT) were a gift from D. Walsh, and Syndecan3 (2E9) was a gift from G. David. Commercially available antibodies were as follows: anti-Psen2-CTF (D30G3) from Cell Signaling, anti-beta-actin (A5441) from Sigma, Lrp1 (EPR3724) from Epitomics, Dcc (A20) and Nrg1 (F-20) from Santa Cruz, and N-cadherin (32/N-cadherin) from BD. For immunohistochemistry, the same App-CTFs were used (1:7,500), and further, NeuN (1:1,000, Chemicon Millipore clone A60 #MAB377, mouse monoclonal), Iba1 (1:900, Wako #019-19741, rabbit polyclonal), and Gfap (1:1,000, Dako #Z0334, rabbit polyclonal). ELISA-capturing antibodies were as follows: JRF/cAb40/28 for Aβ$_{40}$ and JRF/cAb42/26 for Aβ$_{42}$ from Janssen Pharmaceutica (Beerse, Belgium). Detection antibody huAB25-HRPO was obtained from Janssen Pharmaceutica.

## Tissue preparation for morphological evaluation

The mice were sacrificed with a mixture of xylazine (25 mg/ml), ketamine (20 mg/ml) and atropine (20 ng/ml), followed by intracardial perfusion of ice-cold PBS and 4% paraformaldehyde solution. The head was further post-fixed in 4% paraformaldehyde for 48 h at 4°C. The brain was removed, cut in half along the midsagittal plane and processed for paraffin embedding (Thermo Scientific Excelsior™ AS Tissue Processor and HistoStar™ Embedding Workstation). Serial sagittal sections of 6 μm were obtained (Thermo Scientific Microm HM355S microtome). Sections were mounted on Superfrost™ Plus Adhesion slides (Thermo Scientific) and stained with hematoxylin and eosin (H&E, Diapath #C0302 and #C0362) or with antibodies. Sections for App-CTF immunohistochemistry were deparaffinized in xylene and then rehydrated in ethanol series (100, 95 and 70%) and distilled H$_2$O. Endogenous peroxidase was inactivated with 3% H$_2$O$_2$ (15 min, RT). Epitope retrieval was done in citrate buffer (pH 6) using 2100 Retriever. Sections were blocked in 1% BSA solution for 40 min at RT and then incubated overnight at 4°C with the primary antibody followed by 1-h incubation with the secondary antibody. EnVision+/HRP reagent (Dako K400311) was applied on sections for 45 min at RT. Immunoreactivity was revealed with the diaminobenzidine chromogen reaction (Peroxidase substrate kit, DAB, SK-4100; Vector Lab). Slides were counterstained in hematoxylin, dehydrated in ethanol series, cleared in xylene, and permanently mounted with a resinous mounting medium (Micromount Diapath, #60200). 0.1% Tween-20–TBS was used as washing buffer. Immunofluorescence staining for NeuN, Iba1, and Gfap was performed on an automated Ventana Discovery Ultra platform using Alexa Fluor® 568 donkey anti-mouse (1:200, Molecular Probes A10037) or Alexa Fluor® 488 donkey anti-rabbit (1:200, Molecular Probes A-21206) secondary antibodies and DAPI (Sigma-Aldrich D9542) as nuclear counterstain. Imaging and analysis of the bright field and immunofluorescence samples were performed with a Leica DM2500 microscope and a motorized stage-equipped Leica DMRB fluorescence microscope, respectively.

## Analysis of cortical thickness, neuronal density and gliosis

The thickness of parieto-occipital cortex overlying the CA1 hippocampal field was measured in four sagittal H&E-stained brain sections spaced 30 μm apart. The degree of gliosis was determined by means of Gfap and Iba1 immunofluorescence. Gfap- or

**The paper explained**

**Problem**

γ-Secretases are a family of intramembrane cleaving aspartyl proteases and important drug targets in Alzheimer's disease (AD). Several trials with broad-spectrum γ-secretase inhibitors have failed in the clinic, but alternative approaches, for example, inhibitors for selective subtypes of γ-secretase, have been little investigated. Here, we explore the effects of total inhibition of all γ-secretases by combined conditional deletion of the variable *Aph1* subunits in pyramidal neurons of the mice and contrast this with selective deletion of the single subunits.

**Results**

We find a progressive neurodegeneration in the combined *Aph1abc*-γ-secretase *cKO* (conditional knockout) animals. This neurodegeneration is associated with the accumulation of a series of γ-secretase substrates. The cortical atrophy, neuronal loss and gliosis present in *Aph1abc cKO* animals was not modified by deletion of *App* in these animals. We find furthermore that γ-secretase substrates accumulate to a much lesser extent when only *Aph1a* or *Aph1bc* are targeted and that the pattern of accumulation is different between *Aph1a* and *Aph1bc* targeted animals. No neurodegeneration was present in the single targeted animals, while the *Aph1bc* targeted animals display a significant effect on Aβ generation.

**Impact**

In contrast to the hypothesis that the neurodegeneration in conditionally targeted γ-secretase mice is relevant to Alzheimer's disease pathogenesis (loss of function hypothesis), this work indicates that all types of complexes need to be simultaneously inactivated to obtain the neurodegenerative phenotype. This, together with the fact that the phenotype is not modified by deleting *App* expression, suggests that the neurodegeneration in the γ-secretase-deficient mice is not AD related but likely caused by the accumulation of large amounts of hydrophobic protein domains in the neuronal cell membranes and/or the inactivation of crucial γ-secretase-dependent pathways. Furthermore, we demonstrate for the first time the *in vivo* substrate selectivity of two different γ-secretase subtypes.

Iba1-positive areas were measured in two sagittal brain sections spaced 30 μm apart for a total neocortical area of ~3 mm$^2$. The degree of neuronal loss in the neocortex was determined by means of NeuN immunofluorescence. For each animal two sagittal brain sections spaced 30 μm apart were used. Full cortical thickness counts of NeuN-positive cells were assessed in comparable regions of the parieto-occipital cortex. Full cortical thickness counts were then normalized to the total sagittal length of the cortical fields considered in the evaluation (i.e. ~2 mm for each animal). Comparative morphometric analyses were performed on highly homologous and anatomically matched sagittal sections selected based on the recognition of specific neuroanatomic landmarks (Garman *et al*, 2016). The Mouse Allen Brain Atlas (http://atlas.brain-map.org/) was used as neuroanatomic reference. Image analysis was performed using ImageJ software. For each genotype and each time point, three animals were analyzed. Genders were mixed. The morphological evaluation was carried out blindly.

**Tissue preparation for cortical and hippocampal lysates**

Cortices and hippocampi were dissected and homogenized in a Dounce homogenizer in 10 ml/mg tissue of STE buffer (320 mM sucrose, 5 mM Tris–HCl (pH 7.2), 1 mM EGTA) + complete protease inhibitors (Roche). The homogenate was centrifuged at 800 *g* for 10 min at 4°C. 750 μl of the supernatant was further ultracentrifuged at 100,000 *g* for 60 min at 4°C. The pellet (containing synaptosomes as well as membranous organelles such as Golgi, endoplasmic reticulum and plasma membrane) was resuspended in STE buffer + 1% TX-100 and incubated on ice for 30 min. The supernatant was collected after centrifugation at 19,000 *g* for 30 min at 4°C. Equal amounts of protein were taken and analyzed by SDS–PAGE and semi-quantitative Western immunoblot.

**Tissue preparation for Aβ ELISA**

Brains were homogenized in 7.5 volumes of 0.4% diethylamine/50 mM NaCl/1× PI-EDTA solution using FastPrep Lysing Matrix D Tubes in a FastPrep Instrument for 45 s at 6 m/s. Samples were centrifuged for 5 min at 19,000 *g* at 4°C. Supernatant was transferred to prechilled tubes and ultracentrifuged at 250,000 *g* for 30 min at 4°C. Supernatant was neutralized by adding 1/10 volume 0.5 M Tris–HCl (pH 6.8).

**Expanded View** for this article is available online.

**Acknowledgements**

Immunohistochemical analysis was performed in collaboration with Infra-Mouse. We want to thank Annick Francis and Lorna Omodho for assistance with sectioning and staining of the brains. We want to thank Véronique Hendrickx and Jonas Verwaeren for the assistance with breeding the mouse colonies and Annerieke Sierksma for advice with the statistical analysis of the data. The work was supported by the Fonds voor Wetenschappelijk Onderzoek (FWO), the KU Leuven and VIB, a Methusalem grant of the KU Leuven/Flemish Government, and grants from Stichting Alzheimer Onderzoek (SAO-Belgium). HA is a fellow of the IWT, Flanders. BDS is supported by the Bax-Vanluffelen Chair for Alzheimer's Disease and "Opening the Future". This work was supported by Vlaams Initiatief voor Netwerken voor Dementie Onderzoek (VIND, Strategic Basic Research Grant 135043).

**Author contributions**

HA and LS performed experiments, analyzed the data, and wrote the manuscript; ER performed experiments, analyzed the data, and wrote parts of the manuscript; SM and CV provided critical input; EP analyzed the data; UM provided APP knockout mice and analyzed the data; and LC-G and BDS conceived experiments, analyzed the data, and wrote the manuscript. All authors read, provided feedback on, and approved the final version of the manuscript.

**Conflict of interest**

BDS is consultant to Janssen Pharmaceutica and received grants to work on γ-secretase. All other authors declare that they have no conflict of interest.

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
