## [Review Process File · EMBO Molecular Medicine]

Manuscript EMM-2017-07561

Inactivation of γ -secretases leads to accumulation of substrates and non-Alzheimer neurodegeneration

Hermien Acx, Lutgarde Serneels, Enrico Radaelli, Serge Muyldermans, Cécile Vincke, Elise Pepermans, Ulrike Müller, Lucia Chávez-Gutiérrez, Bart De Strooper

Corresponding author: Lucia Chávez-Gutiérrez & Bart De Strooper, Katholieke Universiteit Leuven

Review timeline:

Submission date:	12 January 2017
Editorial Decision:	02 February 2017
Revision received:	25 April 2017
Editorial Decision:	03 May 2017
Revision received:	09 May 2017
Accepted:	11 May 2017

Transaction Report:

Editor: Céline Carret

1st Editorial Decision

02 February 2017

Thank you for the submission of your manuscript to EMBO Molecular Medicine. We have now heard back from the three referees whom we asked to evaluate your manuscript.

You will see from the comments below, that all three referees find the data interesting and convincing and all three are supportive of publication. Still referees 2 and 3 have a few suggestions for addition and comments that we find relevant and would like to encourage you to address. At this stage we will not ask you to perform behavioural analyses/electrophysiology unless you already have some data (which would be great), but a thorough discussion would be nice.

Given the balance of these evaluations, we would welcome the submission of a revised version for further consideration and depending on the nature of the revisions, this may be sent back to the referees for another round of review.

Please let me remind you that it is EMBO Molecular Medicine policy to allow only a single round of revision and that, as acceptance or rejection of the manuscript may depend on another round of review, your responses should be as complete as possible. EMBO Molecular Medicine has a "scooping protection" policy, whereby similar findings that are published by others during review or revision are not a criterion for rejection. Should you decide to submit a revised version, I do ask that

you get in touch after three months if you have not completed it, to update us on the status. Please also contact us as soon as possible if similar work is published elsewhere. If other work is published we may not be able to extend the revision period beyond three months.

Please read below for important editorial formatting, your revised article should conform to our guidelines (authors checklist is missing, Appendix in the wrong format, etc).

I look forward to receiving your revised manuscript.

***** Reviewer's comments *****

Referee #1 (Remarks):

In this manuscript the authors show that combined inactivation of A ϕ 1a/b/c in the forebrain leads to inactivation of gamma-secretase activity followed by aging-dependent cortical atrophy, neuronal loss and gliosis. To test whether accumulation of APP-CTFs (which accumulate in these mice due to inactivation of gamma-secretase activity) is responsible for these phenotypes, the author crossed these conditional A ϕ 1a/b/c KO mice to APP KO mice. The result show that ablating APP expression does not rescue cortical atrophy, neuronal loss and gliosis, suggesting that APP-CTFs are not the main mediators of cortical atrophy, neuronal loss and gliosis. The experiments are well performed and the approach is straightforward and direct.

Referee #2 (Comments on Novelty/Model System):

This is a tour-de-force study using cross of a series of conditional KO mice on gamma-secretase activity, and the data are sound and convincing.

Referee #2 (Remarks):

Acx and colleagues created conditional KO mice lacking A ϕ 1a, b and c in cortical pyramidal neurons by crossing with CaMKII-Cre mice, and observed progressive cortical neuronal loss in a manner reminiscent to those observed in conditional KO mice lacking PS1 and 2. Interestingly, they crossed the A ϕ 1abc conditional KO mice with APP KO mice, but did not see any change in the cortical atrophy, despite the disappearance of accumulation of APP C-terminal fragment, the latter having been implicated in neurodegeneration caused by gamma-secretase ablation. Although the impact of the whole study is somewhat limited because of the lack of the mechanism of neurodegeneration caused by total loss of gamma-secretase activities, the in vivo mouse studies are well designed and the data on the phenotype, especially accumulation patterns of CTFs of different gamma-secretase substrates, are novel and worthy of documentation. The authors may want to clarify a few points regarding methodology and discussions.

(1) In the 3rd paragraph of the discussion, the authors refer to the interesting speculation on relationship between the putative deteriorative phenotype observed in A ϕ 1abc KO mice and AD patients treated with non-selective gamma-secretase inhibitors (eg semagacestat), in relation to the repetitive complete inhibition, (which has been described in Cell by the lead author in Cell in 2014), but this is too speculative and should be more conservatively discussed unless additional evidence had been obtained.

(2) Quantification of neuronal number: the authors quantitated the neuronal "density" rather than the total number of cortical neurons. As frequently cautioned, this may lead to false conclusions, for example by reduction in neuropil volume causing condensation of cell body density; thus, stereologic counting has been strongly recommended in a number of standard neuroscience journals. At least the authors should discuss the relevance of the counting method they used, to support the conclusion that neurons are lost in the cortices of A ϕ 1abc mice.

Referee #3 (Comments on Novelty/Model System):

See comments to authors.

Referee #3 (Remarks):

This, very well done and straightforward paper, builds off previous work in the DeStrooper laboratory where they have suggested that selective targeting of APh1b presenilin complexes should be considered for treatment of AD. Here they conduct complex mouse genetic studies to drive that point home. Using conditional knockouts in forebrain pyramidal neurons they find that single APh1a or bc knockouts have no effect on global phenotypes and only accumulated minimal to modest amounts of various substrates. The selective effect of the KO on different substrates is intriguing and a point of intrigue. The authors then also perform a study in APP KO mice in APh1abc KO and show that the phenotype remains; indicating substrate accumulation (APP CTF) does not drive this phenomenon. This later point strongly refutes some data in the literature that tries to promote CTF accumulation as a driver of AD like neurodegeneration.

There are few issues, mostly in terms of how the findings are discussed, that should be commented on.

1. Lack of behavioral or functional data. Lots of mice show no "degenerative" phenotype but are still not "normal". Given the complexity of the crosses, I am not sure that behavioral data will help (Although some in the field might feel behavioral data is requisite for publication), but perhaps some electrophysiology might be useful. Alternatively, a more balanced discussion about the possibility that subtle problems may not be detected with methods employed.

2. CAMK2 often drives expression in the hippocampus. Are similar degenerative effects noted there?

3. The CNS focus is laudable but a small molecule even if selective for a specific gamma-complex will have systemic exposure. Again some discussion of the fact that inhibition of specific complexes may still have unexpected liabilities is important do not rule this out.

4. A more balanced discussion on this approach vis a vis BACE inhibition should be included. My own "opinion" is that this approach will not again much traction unless BACE inhibitors prove too toxic. The reduction in ABeta by a bc KO is not that impressive and would only be expected to have some benefit in the setting of prevention. SO to me this seems like a backup strategy as we wait for trials on BACE inhibitor safety and efficacy to read out.

Figure 3. Some way to show the MW weights of the various fragments. Right now only 80 kDa is shown and this makes really no sense. Maybe actually provide the calculated MW at which each fragment ran as the full length gels will be part of the data package.

Very Minor stylistic, although very readable there are some times when tense results is altered e.g. we demonstrated is changed to we demonstrate.....be consistent.

1st Revision - authors' response

25 April 2017

Referee #1 (Remarks):

In this manuscript the authors show that combined inactivation of APh1a/b/c in the forebrain leads to inactivation of gamma-secretase activity followed by aging-dependent cortical atrophy, neuronal loss and gliosis. To test whether accumulation of APP-CTFs (which accumulate in these mice due to inactivation of gamma-secretase activity) is responsible for these phenotypes, the author crossed these conditional APh1a/b/c KO mice to APP KO mice. The result show that ablating APP expression does not rescue cortical atrophy, neuronal loss and gliosis, suggesting that APP-CTFs are not the main mediators of cortical atrophy, neuronal loss and gliosis. The experiments are well performed and the approach is straightforward and direct.

We thank the referee for positive and supportive comment.

Referee #2 (Comments on Novelty/Model System):

This is a tour-de-force study using cross of a series of conditional KO mice on gamma-secretase activity, and the data are sound and convincing.

We thank the referee for this positive comment

Referee #2 (Remarks):

Acx and colleagues created conditional KO mice lacking Aph1a, b and c in cortical pyramidal neurons by crossing with CaMKII-Cre mice, and observed progressive cortical neuronal loss in a manner reminiscent to those observed in conditional KO mice lacking PS1 and 2. Interestingly, they crossed the Aph1abc conditional KO mice with APP KO mice, but did not see any change in the cortical atrophy, despite the disappearance of accumulation of APP C-terminal fragment, the latter having been implicated in neurodegeneration caused by gamma-secretase ablation. Although the impact of the whole study is somewhat limited because of the lack of the mechanism of neurodegeneration caused by total loss of gamma-secretase activities, the in vivo mouse studies are well designed and the data on the phenotype, especially accumulation patterns of CTFs of different gamma-secretase substrates, are novel and worthy of documentation. The authors may want to clarify a few points regarding methodology and discussions.

(1) In the 3rd paragraph of the discussion, the authors refer to the interesting speculation on relationship between the putative deteriorative phenotype observed in Aph1abc KO mice and AD patients treated with non-selective gamma-secretase inhibitors (eg semagacestat), in relation to the repetitive complete inhibition, (which has been described in Cell by the lead author in Cell in 2014), but this is too speculative and should be more conservatively discussed unless additional evidence had been obtained.

We have adapted the discussion to indicate that genetic and pharmacological experiments are different. We indicate that this is speculation from our side.

We write: "Although we did not perform pharmacological experiments to investigate this further we speculate that intermittent accumulations of transmembrane domain fragments in neurons might have contributed to the aggravated cognitive performance of the treated patients".

(2) Quantification of neuronal number: the authors quantitated the neuronal "density" rather than the total number of cortical neurons. As frequently cautioned, this may lead to false conclusions, for example by reduction in neuropil volume causing condensation of cell body density; thus, stereologic counting has been strongly recommended in a number of standard neuroscience journals. At least the authors should discuss the relevance of the counting method they used, to support the conclusion that neurons are lost in the cortices of Aph1abc mice.

We agree with the comment of the reviewer. In recent years, it has been argued that 'unbiased' 3-D stereological methods for neuronal counting are more reliable and accurate than 2-D cell-counting methods assessing neuronal density. However, in the present study, it was not possible to implement unbiased 3-D stereological methods using physical disector/fractionator principle as our comprehensive histopathological characterization required the assessment of multiple IHC markers and morphometric parameters on serial paraffin sections obtained from the same brain samples. As a consequence we did not have enough representative material for 3-D stereological studies. Nevertheless, we developed an alternative approach to minimize possible biases associated with 2-D methods and tissue shrinkage in the neocortex:

- In highly comparable regions of the parieto-occipital cortex, we selected large sampling windows where full cortical thickness counts of NeuN-positive cells were performed. This allowed us to consider the neocortex in its complex and non-homogeneous layered structure avoiding application of small and poorly representative sampling windows that are normally used with the physical disector/fractionator principle.*
- We did not express the neuronal counts as cell density (e.g. number of 2-D cell profiles per unit area) but full cortical thickness NeuN+ cell counts were "normalized" to the sagittal cranio-caudal length of the cortical tract selected for the assessment (please note that neuronal density expressed in NeuN+ cells/mm² was erroneously reported in the text and graphs and this has now been replaced by NeuN+ cells/mm of cortical length as clearly was mentioned in M&Ms). This reference to length allowed us to correct for the biases linked to the neuroparenchymal collapse and shrinkage (which is most likely a factor in those groups of mice affected by neurodegeneration).*

We are therefore confident that our approach generated accurate results as significant neuronal loss was observed only in those groups where neurodegeneration was also confirmed through the assessment of other parameters such as neocortical atrophy/collapse and gliosis.

Referee #3 (Comments on Novelty/Model System):

See comments to authors.

Referee #3 (Remarks):

This, very well done and straightforward paper, builds off previous work in the DeStrooper laboratory where they have suggested that selective targeting of APH1b presenilin complexes should be considered for treatment of AD. Here they conduct complex mouse genetic studies to drive that point home. Using conditional knockouts in forebrain pyramidal neurons they find that single APH1a or bc knockouts have no effect on global phenotypes and only accumulated minimal to modest amounts of various substrates. The selective effect of the KO on different substrates is intriguing and a point of intrigue. The authors then also perform a study in APP KO mice in Aph1abc KO and show that the phenotype remains; indicating substrate accumulation (APP CTF) does not drive this phenomenon. This later point strongly refutes some data in the literature that tries to promote CTF accumulation as a driver of AD like neurodegeneration.

We thank the referee for positive comment

There are few issues, mostly in terms of how the findings are discussed, that should be commented on.

1. Lack of behavioral or functional data. Lots of mice show no "degenerative" phenotype but are still not "normal". Given the complexity of the crosses, I am not sure that behavioral data will help (Although some in the field might feel behavioral data is requisite for publication), but perhaps some electrophysiology might be useful. Alternatively, a more balanced discussion about the possibility that subtle problems may not be detected with methods employed.

The referee is correct and we mention this point now clearly in the discussion

2. CAMK2 often drives expression in the hippocampus. Are similar degenerative effects noted there?

The effects in the hippocampus are much less dramatic. We mention this briefly in the discussion. The problem is that negative data are very difficult to interpret here. Is this because of less efficient Cre recombination in the hippocampus for instance, or does it really indicate resilience of this brain area for the type of neurodegeneration we see here. We have not studied this aspect in further detail.

3. The CNS focus is laudable but a small molecule even if selective for a specific gamma-complex will have systemic exposure. Again some discussion of the fact that inhibition of specific complexes may still have unexpected liabilities is important. Do not rule this out.

We agree fully with this point, the APH1b/c full knock out phenotype has been described in our previous publication in Science (Serneels et al, 2009) and is very mild. But we agree with the referee that we should indicate this point better in the discussion. We have added:

“A word of caution is indicated however. First, we analyzed mice only at biochemical and microscopic-morphological levels, and we can thus not make any statements with regard to potential behavioral or electrophysiological abnormalities in the different mice investigated here. Second, it should be stressed that we here only inactivate γ -secretases in neurons that express Cre driven by the CAMK2 promoter. Pharmacological inhibitors would block also γ -secretases in other cells and peripheral tissues and therefore further scrutiny is needed. We previously generated full Aph1b/c knock outs and have shown that these mice are healthy and show very little problems apart from a deficit in operational memory (Serneels et al., 2009; Dejaegere et al., 2008).”

4. A more balanced discussion on this approach vis a vis BACE inhibition should be included. My own "opinion" is that this approach will not gain much traction unless BACE inhibitors prove too toxic. The reduction in ABeta by a bC KO is not that impressive and would only be expected to have some benefit in the setting of prevention. SO to me this seems like a backup strategy as we wait for trials on BACE inhibitor safety and efficacy to read out.

We agree, also in the light of the failure of verubecestat and other amyloid directed therapies we will have to see what (if any) the place of anti-amyloid drugs in Alzheimer treatment will be. However we do not think that this manuscript is the best place to discuss this problem.

Figure 3. Some way to show the MW weights of the various fragments. Right now only 80 kDa is shown and this makes really no sense. Maybe actually provide the calculated MW at which each fragment ran as the full length gels will be part of the data package.

We have added molecular weight markers and updated the figure.

Very Minor stylistic, although very readable there are some times when tense results is altered e.g. we demonstrated is changed to we demonstrate.....be consistent

We have carefully checked the correct use of tenses, we hope that this is now improved.

We have consulted Annerieke Sierksma as advisor on the statistical analysis of our data. Reanalysing the data according to her recommendations resulted in some minor changes but those changes aren't affecting the general conclusions. Changes are now included in the main text, figures and legends.

We changed some minor stylistics;

All mouse proteins are presented as "only first capital letter symbols".

All human proteins are denoted with all capital letter symbols.

CamKIIa is consequently spelled in the same way.

2nd Editorial Decision

03 May 2017

Thank you for the submission of your revised manuscript to EMBO Molecular Medicine. We have now received the enclosed reports from the referees that were asked to re-assess it. As you will see the reviewers are now supportive and I am pleased to inform you that we will be able to accept your manuscript pending the following final editorial amendments:

- 1) Please carefully check the authors guidelines for formatting your supplemental information: Expanded view and Appendix (see: <http://embomolmed.embopress.org/authorguide#expandedview>)

The most simple way would be to reliable all Supplementary figures as EV figures and update the callouts as Figure EV1 and so on. EV figure legends must remain in the main article.

Please submit your revised manuscript within two weeks. I look forward to seeing a revised form of your manuscript as soon as possible.

***** Reviewer's comments *****

Referee #2 (Comments on Novelty/Model System):

The authors have improved the technical accuracy of the data on neuronal number, by adding an alternative quantitation method, and also revised the discussion in a more persuasive manner. Now it can be published in this present form.

Referee #2 (Remarks):

The authors have now addressed all my comments in a suitable fashion.

Referee #3 (Comments on Novelty/Model System):

An elegant use of mouse genetic models

Referee #3 (Remarks):

The authors more than adequately addressed my concerns

Corresponding Author Name: Bart De Strooper and Lucia Chavez-Gutierrez

Manuscript Number: EMM-2017-07561